# Dialects of Madagascar

**Maurizio Serva, Michele Pasquini**[ID]*

Dipartimento di Ingegneria e Scienze dell'Informazione e Matematica, Università dell'Aquila, L'Aquila, Italy

* michele.pasquini@gmail.com

## Abstract

All results in this paper are based upon a new dataset consisting in 60 Swadesh lists of 207 items, overall 12,420 terms collected during 2018-2019. Each list corresponds to a different variety of Malagasy, which is not simply identified by the name of the ethnicity but also by the precise location where the variety was collected. This is very important since some traditional ethnic groups are a heritage of historical events rather than representing communities with similar habits and dialects. This new dataset is by far the best available, both for dimension and completeness. The varieties are classified both by standard tools, as the trees generated by UPGMA and NJ which privilege genealogy by detecting vertical transmissions, and by a new method which privileges horizontal exchanges. The new method results in a two-dimensional chart of Madagascar which realistically reproduces geography despite being generated only by comparison of words. The landing date of the ancestors of Malagasy is determined about 650 *CE*. This result is obtained by a straightforward approach based on the comparison of the UPGMA Malagasy family tree with the analogous tree of Romance family of languages for which all dates are well historically attested. We also propose an improved definition of Diversity computed for every locus in Madagascar and not only in places where the dialects were collected. Moreover, Diversity becomes a locally determined quantity as it is usually in biology. Diversity differences point to the South-East coast as the location where the first colonizers landed or, at least, where Malagasy variants started their dispersion. Finally, we find that the dialect spoken by the Mikea, a hunter-gatherer people in the South-West of Madagascar, is not very different from the variants of their neighbours Vezo and Masikoro. Therefore, Mikea unlikely can be linked to eventual aboriginal populations living in Madagascar prior to the main colonization event in 650 *CE*.

## 1 Linguistic, genetic and archaeological preamble

The Austronesian expansion, which very likely started from Taiwan or from the South of China, is probably the most spectacular event of maritime colonization in human history. There is a huge literature on this subject, let us just mention a very recent survey [1] with an interesting hypothesis concerning the correlation between expansion bursts and technological innovations.

**Data Availability Statement:** All relevant data are within the paper and its Supporting Information files.

**Funding:** MP; grant "Fisica Matematica: Realizzazione di applicativi simbolici per l'elaborazione di database testuali", Prot. n. 2949, 27 August 2018, Repertorio n. 194, Università degli

Studi dell'Aquila; https://www.univaq.it/; The funder had no role in study design, data collection and analysis, decision to publish, or preparation of the manuscript.

**Competing interests:** The authors have declared that no competing interests exist.

Madagascar is the Western edge of this expansion, in fact, although the genetic makeup of Malagasy people is African and Indonesian with comparable proportions, the Malagasy language and its variants belong to the Austronesian family.

## 1.1 Linguistics

The Dutch merchant Frederick de Houtman van Gouda was the first to notice that the Malagasy natives speak a language "very similar to Malay" [2]. A decade later the Portuguese Jesuit Luis Mariano arrived to the identical conclusion noticing a similarity between the Malagasy speech and the Malay languages [3]. Subsequently, the work of Herman Neubronner van der Tuuk [4] established beyond doubt the relationship between Malagasy and other Indonesian languages (but he wrongly proposed a close relationship with Toba Batak).

Finally, the incontestable link to a precise Indonesian language is due to the Norwegian missionary Otto Christian Dahl (1903-1995) who begun his apostolic mission in Madagascar in 1929 and later, in 1935, embarked on linguistic studies (a short bibliography in [5]). His missionary vocation and his interests in linguistics immediately found common ground [6] and some years later he published his fundamental work [7] where he firmly established a striking kinship between Maanyan, spoken in the South-East of Kalimantan, and Malagasy (see also [8] and [9]). His main conclusion was summarized by himself as: "Le Maanjan et le malgache sont des languages très etroitement apparentées. Parmi les langues indonésiennes connues actuellement aucune n'a autant de ressemblances avec le malgache que le Maanyan" [7]. This statement after 70 years is still undisputed.

In his 1951 paper Dahl also proposed, on the basis of historical and linguistic evidence, that the date of the landing event was 400 *CE*. His dating was shortly confirmed by Isidore Dyen who introduced the lexicostatistics grouping and dating methods into the Malagasy debate [10].

In [11], for the first time, it was considered that Malagasy should not be considered a single language but a constellation of dialects well different each other. Subsequently, by considering Swadesh 100 words list of sixteen different varieties, the authors of [12] were able to perform a lexicostatistical and glottochronological study concluding that the most diverse dialects were those in the North and that the Island was first colonized around two thousand years ago. These possibly wrong conclusions were a consequence of the fact that the research was performed without the help of the modern algorithms for building trees, the analysis of their data by recent tools brings to different conclusions [13]. The Vérin *et al.* dating for the landing event was confirmed in [14] on the basis of archaeological findings, but in this second paper it is admitted that the remains cannot be attributed with certitude to the ancestors of modern Malagasy.

The discovery of the Maanyan connection correctly collocated Malagasy at its place among East-Barito languages (so called after A. B. Hudson [15]), but it did not resolve the relation with the other Austronesian languages. Dahl himself addressed this problem [8] and occasionally traced Malagasy words to Ngaju [7, 9], but the first systematic study of non-Maanyan connections was performed by Alexander Adelaar. This author first considered the relations with Malay [16], and later also Javanese, Ngaju and Malay variants as Banjarese [17–19].

A careful timing of borrowings, which is derived from the known history of the Indian Ocean, allowed Adelaar to state that the founding event occurred between 600 *CE* and 700 *CE*. In [16] he wrote "Dahl (1951) used the presence (in his counting) of 30 Sanskrit loanwords in Malagasy as evidence that the migrations to East Africa must have taken place after the

introduction of Indian influence in Indonesia. The oldest written evidence of Indian presence in this area is a Sanskrit inscription from around 400 AD found in Kutai, South East Borneo. Dahl therefore proposed the 5th century AD as the most likely migration period. In my 1989 article I interpret the Sanksrit influence in a different way. All but one of the Sanksrit loan-words in Malagasy have corresponding forms in Malay and/or Javanese. Moreover, many Sanskrit loanwords show the same phonological adaptations as their Malay and Javanese counterparts; I therefore conclude that the loanwords in question were actually borrowed via Malay or Javanese. As a consequence, the migration date should not be correlated to the beginning of Indianisation in the archipelago, but more specifically to the emergence of Sanskrit influence on Malay. This influence was the manifestation of an Indian Malay civilization, which was evidenced for the first time in the emergence of the maritime polity of Srivijaya in the seventh century AD in South Sumatra".

Adelaar also remarked that "Malay loanwords in Malagasy often pertain to a maritime environment (which includes names of winds and directional terms)" and that the Maanyan speakers live along the rivers of Kalimantan, hence whithout having the necessary skills for long-distance maritime navigation. He proposed as a possible explanation that the Maanyan speakers constituted the crew of expeditions led by Malay sailors.

This is not the end of the story, since Adelaar also provided some additional evidence that a number of words in Samihim (a Barito language), which do not occur in Maanyan, have cognates in Malagasy [20]. Moreover, in [21] he highlighted the relevance of lexical borrowings from South Sulawesi languages, while rejecting the hypothesis of borrowings from Philippine proposed in [22].

To complete this unexpected complex picture of the lexical sources of Malagasy, Adelaar also described the influences of Bantu languages [23]. Indeed, this topic was already addressed in 1954 by Dahl, which wrote about the influence of Comorian Bantu over Malagasy especially in phonology [24], and later by Blench and Walsh in [25, 26] and [27], which focused on domesticated animals and mammals names. Philippe Beaujard [28] argued that the first migrants Indonesian came into contact with Bantu speakers only after their arrival in Madagascar, probably at the end of the first millennium *CE*.

Some topics discussed above were also considered in [13] (see also [29]) through the application of new quantitative methodologies inspired by, but nevertheless different from, classical lexicostatistics and glottochronology. All approaches in these papers converge to the conclusion that Malagasy dialects are classified into two main geographical subfamilies: South-West and Center-North-East. Moreover, a date of landing was determined around 650 *CE*, in agreement with the proposition made many years before by Adelaar [16]. Finally, by means of a technique which is based on the calculation of differences in linguistic Diversity proposed in [30] and which is a quantitative implementation of a well known argument in linguistics and biology [31, 32], it was argued that the landing took place on the South-East coast of the Island. The South-Eastern landing was also proposed by various historians and anthropologists (see, for example, [33]).

Adelaar in [34] refutes the hypothesis in [13], and argues that the South-East landing ".. is not supported, as it concerns later migrations, taking place half a millennium after the first SEB speakers had arrived. If anything, these migrations must have a skewing effect on Serva *et al*.'s outcomes and on Sapir's principle, rather than showing the place where SEB speakers first arrived and from where they dispersed all over Madagascar.".

On the basis of both linguistic data and Geo-maritime arguments, other scholars arrived to the conclusion that the colonization started from North. This Northern hypothesis is clearly motivated in the passionate writings of Rory Van Tuyl [35].

## 1.2 Genetics

Although recent work in linguistics has highlighted the non-Barito contribution to the Malagasy, there is still unanimous consensus about its collocation among East-Barito languages. This view seems in contrast with recent findings of genetic research.

In [36] it is reported that the mtDNA Polynesian motif, which can be found in Malagasy individuals at a macroscopic rate, is found at a low rate in Borneo, not at all in the Barito river area. At higher rate it is found in Sulawesi and coastal Papua New Guinea, and it is predominant in Polynesia. Nevertheless, it is unlikely that this motif came to Madagascar from Polynesia by some direct genetic contact because Polynesian Y chromosome haplogroups have not been found among Malagasy paternal lineages [37]; more likely it came from Borneo, but not directly from Maanyan speakers ancestors.

In [38] and [39] it is shown that both maternal and paternal lineages of Malagasy point to multiple regional sources in Indonesia, with a focus on Southern and Eastern Borneo, but also Sulawesi and the Lesser Sunda islands. Surprisingly, the Malagasy do not exhibit a clear genetic connection with the Maanyan, despite the obvious linguistic association. The suggested explanation is that the "settlement may have been mediated by ancient sea nomad movements". This opinion is also shared by [40] which also proposes that "the distribution of a musical instrument such as the valiha among the highland Merina, points to some sort of direct Sulawesi connection". This may be true, but we would not attribute as much value to shared musical instruments as to linguistic evidence.

Finally, [41] found strong "support for an origin of the Asian ancestry of Malagasy among the Banjar. This group emerged from the long-standing presence of a Malay Empire trading post in South-East Borneo, which favored admixture between the Malay and an autochthonous Borneo group, the Maanyan". The present day Banjar people speak a Malay language, but the authors argue that the ancestors of the Banjar spoke a language close to the reconstructed Proto-Malagasy, in turn, forcefully close to nowadays Maanyan.

For what concerns the African component of Malagasy genetic makeup, it is well known that it is approximately of the same size of the Asian component [42]. Nevertheless, in [43] it was shown that "the distribution of ancestral components was ethnic and sex biased, with the Asian ancestry appearing more conserved in the female than in the male gene pool and in inland than in coastal groups". Similar conclusions concerning the African-Indonesian admixture were reached in [44], leading to important consequences: "The distribution of African and Asian ancestry across the island reveals that the admixture was sex biased and happened heterogeneously across Madagascar, suggesting independent colonization of Madagascar from Africa and Asia rather than settlement by an already admixed population. In addition, there are geographic influences on the present genomic diversity, independent of the admixture, showing that a few centuries is sufficient to produce detectable genetic structure in human populations." On the other side, it should be also considered that the sex bias could be seen in the light of kinship structure, which is matrilineal. The same matrilineal (and not patrilineal) preponderance of Austronesian DNA is seen in the East of Indonesia [45].

Very recent research [46] found evidence of a weak but well detectable Malagasy genetic flow in Somalia and Yemen, which supports the existence of African-Malagasy contacts after the main Asian colonization event of the Island.

Finally, an extremely interesting contribution to the understanding of the colonization of Madagascar comes from [47], where it is proposed a "scenario in which Madagascar was settled approximately 1200 years ago by a very small group of women (approx. 30), most of Indonesian descent (approx. 93%). This highly restricted founding population raises the possibility

that Madagascar was settled not as a large-scale planned colonization event from Indonesia, but rather through a small, perhaps even unintended, transoceanic crossing."

### 1.3 Archaeology

One of the questions which still await a definitive answer is: there were someone around before the arrival of the ancestors of nowadays Malagasy?

A clear answer will hardly be provided by linguistics, it may be that some hint will come from genetics, but in principle only archaeology allows going back sufficiently deep into the past, unfortunately this discipline provides contrasting evidence.

In [48], it is reported that a sediment core from Lake Kavitaha, central Madagascar, provides a stratigraphic record showing a marked rise in charcoal about 1300 year *BP* followed by a decline in pollen of woody taxa, culminating in a change to grass-dominated pollen spectra within about 4 centuries. In a much more recent paper ([49]), it is shown that starting from 890 *CE* carbon stable isotopic data indicate a rapid, complete transformation from an open forest landscape to grassland system as a result of the impact that early inhabitants had on the environment. Other recent research ([50]) suggests that grassland formations on Madagascar are natural although it is admitted that it is possible that human disturbance may have resulted in a much larger modern extent of grassland than in pre-human settlement of Madagascar.

Similar dating is found out in [51]: "At Nosy Mangabe we have an apparently continuous occupation dating to the late eighth century [...]. At present, there is no direct evidence for occupation of the interior during the first millennium *CE*".

All dating in the above mentioned papers are compatible with a landing event in seventh century as proposed by prevalent linguistic research.

Nevertheless, in other research ([52]) it is claimed that the minimum age for initial human presence on the island may be set at approximately 2000 *BP*, on the basis of AMS 14C dates for human-modified femora of extinct dwarf hippos from South-West Madagascar. In more recent research [53], dating is further pushed to the past: "Multiple lines of evidence point to the earliest human presence at ca. 2300 years *BP*", a dating which is also embraced in [54]. Moreover, in [55] it is reported that Lakaton'i Anja near Antsiranana, in the North, yielded several stratified assemblages that indicate occupation from at least 4000 years *BP*.

This race to the past seems to progress unabated, in fact, the authors of [56] report human-modified bones dating before 10500 *BP* predating all other evidence. This dating suggests prolonged human-faunal coexistence with limited bio-diversity loss. In [57], the authors underline that recent contributions yielded estimates of initial settlement which may differ as much as 9000 years. Their assessment is that the presence of people on Madagascar dates at least 2000 years *BP*, but also that an Early Holocene arrival cannot be rejected.

All these further dating cannot correspond to the Indonesian colonization performed by the ancestors of modern Malagasy but it could go back to previous inhabitants of Madagascar. This argument will be examined in more details in Section 3.

Some cautious rethinking about human presence on the island can be found in [58], where the authors underline the nonexistence of butchery marks prior of about 1200 years *BP*. Moreover, their close analysis of the Lakaton'i Anja chronology suggests the site dates 1500 years *BP* or more recent, while older dating is due to extensive bioturbation at the site [59]. According to the authors, these findings indicate initial human colonization of Madagascar 1350–1100 years *BP*.

An even more decided criticism to early presence of humans in Madagascar can be found in [60], where the author concludes that there is no compelling evidence that people were present on Madagascar before the mid-first millennium *CE*. All claims and proofs contrasting this

conclusion are re-examined in detail and rejected on the basis of general criteria of archaeological validation.

This last paper [60] and the previous two [58, 59] lead again to a dating which fits the linguistics one for the main colonization event.

## 1.4 Preamble summary and the contributions of this paper

We have seen that much has been understood about the colonization of Madagascar, and also that much has yet to be unveiled.

Consensus seems to converge on the hypothesis of an arrival in the second half of first millennium *CE*, probably in seventh century. In this paper we confirm this dating by a new but simple argument. We assume people arrived by direct navigation from Indonesia, while Africans' contribution to the Malagasy genetic pool and, at a much lesser extent, to the Malagasy language, took place at a later time.

The colonizers from Indonesia were a quite small and possibly heterogeneous group of people, mostly from South-East Borneo. Likely a large part of them were the common ancestors of Malagasy and Banjar people of Borneo. Banjar speak now a Malay variety, but it is argued that their ancestors spoke an East-Baritio language related to the modern Maanyan language, which is the closer language to Malagasy.

After arrival, the language started a process of diversification which led to modern varieties. The relationships among these varieties probably reflect the historical process of internal colonization. In this paper we produce cladograms by standard tools as UPGMA and NJ, but also we propose a new method of classification which better accounts for the fact that in a strongly inter-related network of varieties a simple genealogical description of language relations is inadequate.

We also propose that the colonization started from the South-East of the Island adapting a simple argument from linguistics (and biology) which identifies the location of maximum linguistic Diversity with the homeland of a family of languages. To reach this goal we improved the definition of Diversity given in [13], in order that it is defined in all geographical locations and not only in those places where varieties were collected.

The South-Eastern landing is compatible with a direct navigation from Indonesia, so that admixture with East-African may have taken place after the main Asian colonization event [28].

A very animated discussion among scholars is the eventuality that Madagascar was inhabited before it was colonized by the ancestors of modern Malagasy. The opinions are very different, ranging from proposals of an early occupation thousands of years *BP* to vigorous rejection of any human presence before the main colonization event in the second half of first millennium. It has been also argued that hunter-gatherer populations could show traces of an old component in their language different from modern Malagasy language. We don't find any trace of this component in Mikea hunter-gatherers, who are often suggested being living descendants of the pre-existing inhabitants; more likely they represent a former neolithic people which went back to hunting and gathering for historical reasons. Nevertheless, we cannot exclude that an early occupation occurred leaving still undetected linguistic traces or, eventually, leaving no traces.

## 2 Linguistic data and distances between languages

The dataset we use in this paper consists in 60 Swadesh lists of 207 items, overall 12,420 terms collected by one of us (M.S.) during the years 2018 and 2019. Each list corresponds to a different variety, which is not simply identified by the name of the ethnicity but also by the location

where the variety was collected. In turn, the location is identified by the name of a town/village and by latitude and longitude.

This is very important since some traditional ethnic groups are a heritage of historical events (or a legacy of the colonial *divide et impera* policy) rather than representing communities with similar habits and dialects. In fact, the various isoglosses of a given ethnicity can be extremely different one from the other. For example, some of the varieties of the Sakalava ethnicity belong to the Northern branch of Malagasy dialects and others to the Southern one.

Even recent events resulted in a speech diversification. For example the displacement of 80,000 Antanosy individuals in the Onilahy river valley to escape Merina rule about 1850 *CE* gave rise to a variety different from Antanosy variety spoken in Tolagnaro [61].

That is why varieties cannot be simply identified by the name of the ethnicity, but it is essential to specify the location. In turn, the mere location is not sufficient, since different varieties may cohabit in the same village/town. This simple but important truth was substantially ignored in other research, allowing a certain degree of confusion.

Each list was furnished and checked at least by three native language speakers which, for each given meaning, were asked to furnish the most common word in their dialect as spoken in their town/village.

A complete overview of dialect's geographical locations is given in Fig 1, the colors correspond to the classification which we propose in next section (for the moment the reader can ignore them). Both in Toliara and Morondava two different varieties coexist. The names of the ethnicities are missing in Fig 1 in order to keep it readable, nevertheless they can be deduced by comparison with Fig 2. Eventually, the precise latitude and longitude coordinates as well the names of the corresponding towns/villages and ethnicities can be found in S1 Table.

The complete dataset of 207 items Swadesh lists for 60 Malagasy variants in text format can be found in S1 Dataset. The total number of terms is 12,420.

In order to obtain a dating for the main landing event of the Asiatic colonizers, we will perform a comparison with the well attested historical evolution of Romance languages from Latin, therefore, we also need a relative dataset. Our main source is "The Global Lexicostatistical Database, Indo-European family: Romance group" (September 2016), which contains annotated Swadesh lists compiled by Mikhail Saenko. This dataset is a part of "The Global Lexicostatistical Database" which can be consulted at http://starling.rinet.ru/new100/main.htm. This dataset was implemented by the authors of the present paper as described in [62].

The large amount of information contained in the 60 short vocabularies of our dataset, for the purposes of this paper, is partially encoded into the $60 \times 59/2 = 1770$ distances between each pair of languages. This absolutely does not mean that all or most of the information is transferred into the matrix of distances, we are well aware that in the operation much of it is lost. However, as it has been said, the entire 12,420 items dataset can be consulted in S1 Dataset and freely used, as long as its origin is quoted.

The idea of measuring relationships among languages using vocabulary is much older than modern lexicostatistics, and it seems to have its roots in the work of the French explorer Jules-Sébastien-César Dumont d'Urville. He collected comparative word lists during his voyages aboard the Astrolabe from 1826 to 1829 and, in his work about the geographical division of the Pacific [63], he proposed a method to measure the degree of relation among languages. He used a core vocabulary of 115 terms, then he assigned a distance from 0 to 1 to any pair of words with the same meaning and finally he was able to determine the degree of relation between any pair of languages.

About 70 years ago Morris Swadesh proposed a very similar method [64] using core vocabularies with 100 or 200 terms. Each of these vocabularies (Swadesh lists) contain the words associated to the same *M* meanings (the original Swadesh choice was $M = 200$), which refer to

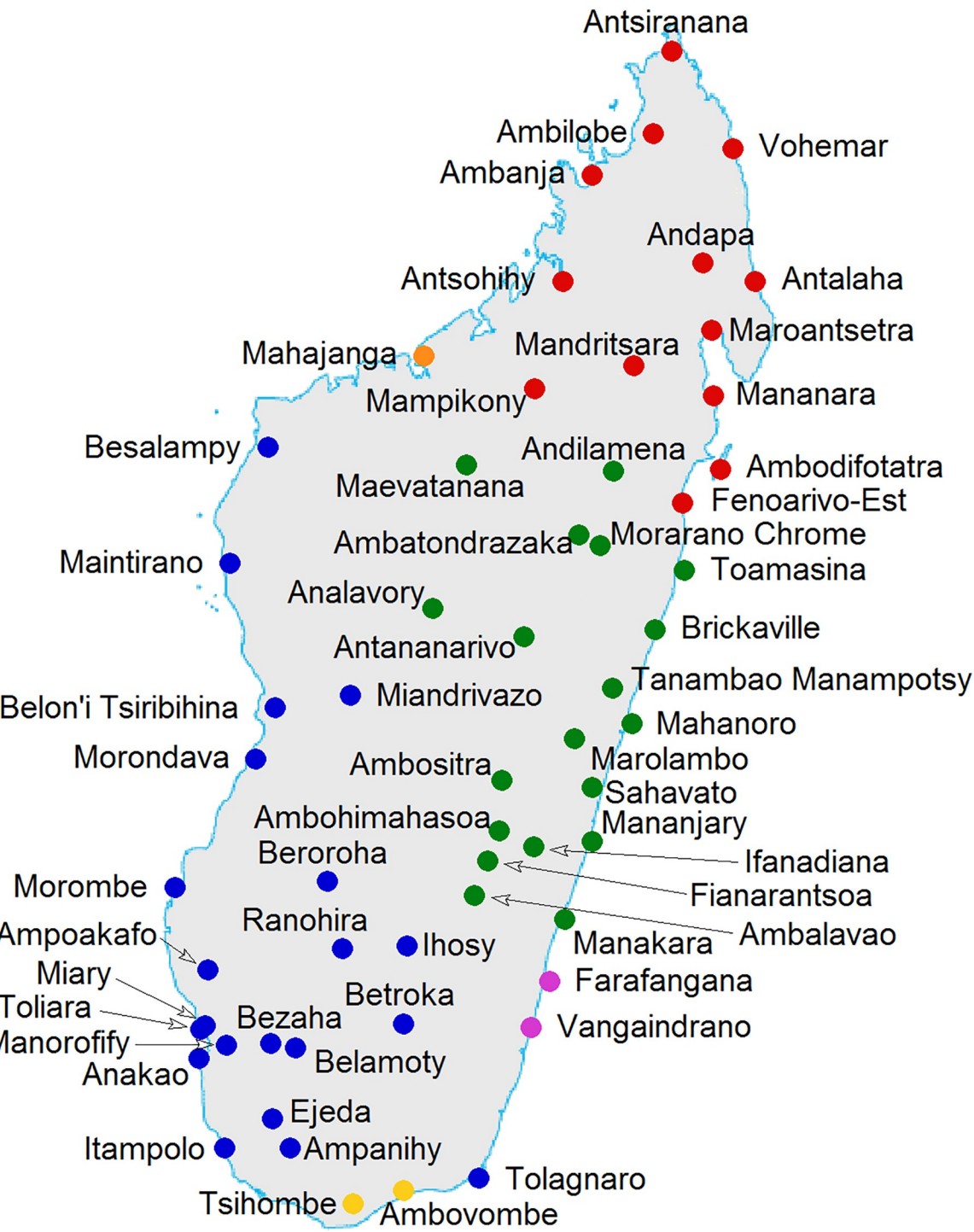

**Fig 1. The map with the names of the towns/villages where each variety was collected.** The names of the ethnicities are missing, nevertheless they can be deduced by comparison with Fig 2. Eventually, the precise latitude and longitude coordinates as well the names of the corresponding towns/villages and ethnicities can be found in S1 Table. Colors correspond to the classification proposed in Section 3. Both in Toliara and Morondava two different varieties coexist, but in Morondava they are close each other (both are blue), while in Toliara they are distant (one is yellow and unreported, the other is blue).

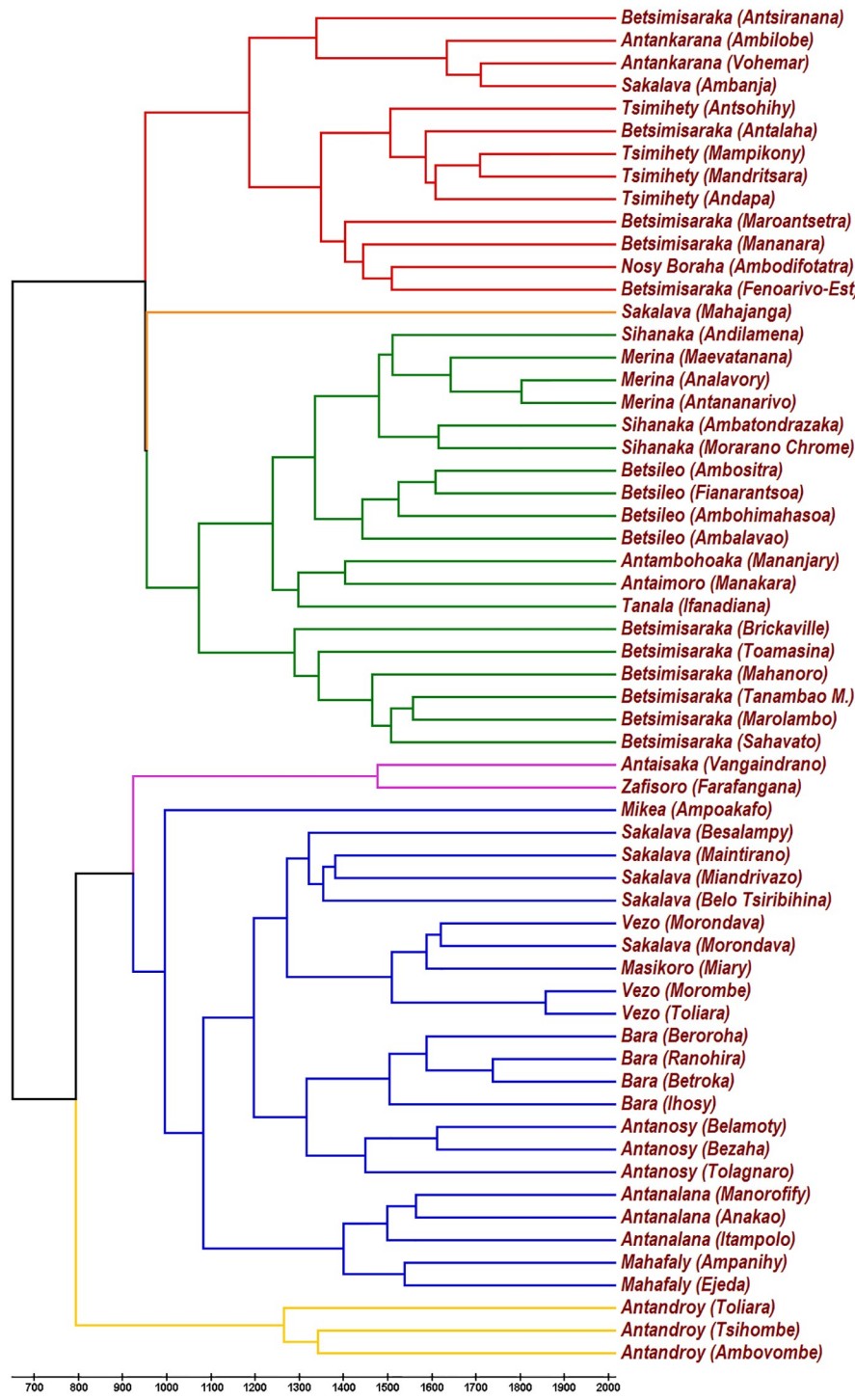

**Fig 2. UPGMA tree obtained from the genealogical distances $T(\alpha, \beta) = -\frac{\tau}{2} \ln C(\alpha, \beta)$.** The value of the characteristic time $\tau$ is chosen between 5027 and 5245 years according to the analysis of subsection 3.2. This choice implies a distance from the root to leaves between 1338 and 1396 years. Accordingly, the founding event (landing of the Asian ancestors and relative start of the diversification of varieties) remains fixed around 650 *CE*.

the basic activities of humans. Comparing the two lists corresponding to a pair of languages it is possible to determine the percentage of cognate pairs (pairs of words with same meaning and a common ancestor word), which, in principle, can be used as a measure of their degree of relation.

Lexicostatistics had seesawing fortunes after Swadesh seminal paper, but over the last 30 years we have seen an increasing acceptance of its methods. The scope of the theory has been enlarged, and new applications have been found [65–67]. Moreover, applications to specific problems have increasingly shown that the methodology invented by Swadesh can be effective to enlighten specific problems concerning the genealogy of languages [68–71].

The strategy of lexicostatistics for finding a distance between any pair of languages can be stated as follows. First, any item of a list is labeled by the index $i$ with $i = 1, 2, \ldots, M$, where $M$ is the number of items in each list (207 in our case), then any language is labeled by the index $\alpha$ with $\alpha = 1, 2, \ldots, N$, where $N$ is the total number of languages (or dialects). Therefore, $\alpha_i$ represents the word corresponding to the item $i$ in the language $\alpha$, which implies that $\alpha_i$ indicates a couple of coordinates, $i.e.$, $\alpha_i = (\alpha, i)$. Then, the distance between words $\alpha_i$ and $\beta_i$ (same meaning, different languages) is defined as

$$D(\alpha_i, \beta_i) = \begin{cases} 0 & \text{if } \alpha_i \text{ and } \beta_i \text{ are cognate,} \\ 1 & \text{otherwise.} \end{cases} \tag{1}$$

The lexical distance between languages $\alpha$ and $\beta$ is then derived averaging over all meanings, $i.e.$,

$$D(\alpha, \beta) = \frac{1}{M} \sum_{i=1}^{M} D(\alpha_i, \beta_i), \tag{2}$$

which is, by definition, the proportion of negative cognate matching and it is obviously a number between 0 (all pairs $\alpha_i$ and $\beta_i$ of terms with same meanings in the two languages have a common ancestor) and 1 (all pairs $\alpha_i$ and $\beta_i$ of terms with same meanings in the two languages have no common ancestor). The overlap $C(\alpha, \beta) = 1 - D(\alpha, \beta)$, on the contrary, is the proportion of positive cognate matching. The number of distance values obviously equals the possible pairs of languages $N \times (N - 1)/2$ (1770 in our case).

The fundamental formula of Glottochronology states that the genealogical distance of two contemporary languages (time from the last common ancestor language) is

$$T(\alpha, \beta) = -\frac{\tau}{2} \ln C(\alpha, \beta), \tag{3}$$

where the characteristic time $\tau$ is the inverse of the replacement rate.

In the idea of Swadesh $\tau$ should have been a universal constant, but this is not the case for many reasons, as the incidence of horizontal transfers (see for example [72]), and different replacement rates for different meaning or different language families (see for example [62]).

To overcome these difficulties Swadesh himself proposed for $\tau$ a value smaller than the inverse of the replacement rate but without a clear recipe for fixing its actual value. In principle one can fix $\tau$ by history. In fact, a single event, which fixes the time from the last common ancestor of a single pair of languages, is sufficient to determine $\tau$ and this value can be hopefully extended to all pairs of languages in the same family. For example, Iceland was colonized by Norwegians about the year 900 $CE$ and therefore the corresponding genealogical distance between Norwegian and Icelandic is about 1.1 millennia. Given that, the overlap between Icelandic and Norwegian can be easily computed, $\tau$ remains determined.

In next section we will describe a strategy for determining $\tau$ which is also based on history, but it relies on a kind of average genealogical distance of the members of a family from their last common ancestor and, therefore, it is in principle more trustworthy.

It is worth to mention that the present work uses an automated version of the Swadesh approach which was proposed about ten years ago [73]. The definition (1) was replaced by a more objective measure based on a normalized Levenshtein distance (*NLD*).

Given two words $\alpha_i$ and $\beta_i$ corresponding to the same meaning $i$ in two languages $\alpha$ and $\beta$, their normalized Levenshtein distance $D(\alpha_i, \beta_i)$ is

$$D(\alpha_i, \beta_i) = \frac{D_L(\alpha_i, \beta_i)}{L(\alpha_i, \beta_i)}, \tag{4}$$

where $D_L(\alpha_i, \beta_i)$ is the Levenshtein distance between the two words and $L(\alpha_i, \beta_i)$ is the number of characters of the longer of the two. This normalized Levenshtein distance, which can take any rational value between 0 and 1 replaces (1), while (2), (3) and the definition of the overlap $C(\alpha, \beta) = 1 - D(\alpha, \beta)$ are left unchanged.

This strategy enormously economizes working time and for some respect it is more objective and reliable. Anyway, various test we made in the last years, and also tests concerning the present work, lead to the conclusion that all results remain the same if standard cognate counting is used instead.

The $N \times N$ upper triangular matrix whose entries of the matrix are the $N(N - 1)/2 = 1770$ lexical distances $D(\alpha, \beta)$ between all pairs of languages is contained in S2 Table.

## 3 Trees and a date for the landing event

The sole ingredient in this section is the upper $N \times N$ upper triangular matrix whose entries are the $N(N - 1)/2 = 1770$ lexical distances $D(\alpha, \beta)$ between all pairs of languages. The matrix of lexical distances $D(\alpha, \beta)$ is transformed into a matrix of genealogical distances $T(\alpha, \beta)$ according to formula (3) with $C(\alpha, \beta) = 1 - D(\alpha, \beta)$.

### 3.1 Classification of Malagasy dialects

The information concerning the vertical transmission of vocabulary from Proto-Malagasy to the contemporary dialects can be extracted from the matrix of genealogical distances by a phylogenetic approach. There are various possible choices for the algorithm for the reconstruction of the family tree, we show the tree generated by Unweighted Pair Group Method Average (UPGMA) and by Neighbor Joining (NJ).

In Fig 2 we report the UPGMA output. The leaves of the tree are identified by the name of the dialect which is followed by the name of the town/village where it was collected and whose location can be appreciated in Fig 1. The absolute time-scale (*i.e.*, the value of $\tau$ in (3)) is calibrated by the results of next subsection, which indicates the date 650 *CE* for the start of diversification of Malagasy varieties. Nevertheless, the timescale it is irrelevant for discussion in this section since the shape of a tree remains unchanged when $\tau$ is modified.

The UPGMA tree shows a main partition of Malagasy dialects into two main branches (Center-North-East and South-West), this partition coincides with the one in [13] but it is at variance with the main partition of a previous study [12]. The difference is a consequence of the fact that the second paper [12] was written without the help of the modern algorithms for building trees, the treatment of their data by recent tools gives the same main partition which is found in this paper (see [13] for a discussion of this point).

In turn, each of the two branches splits into two sub-branches. The South-West branch, whose leaves are associated to yellow, blue and violet, splits in a large blue-violet sub-branch

and a smaller yellow one for Antandroy dialects which show a remarkable isolation. It should be considered that part of the divergence of Antandroy dialects can be explained by the common/polite dichotomy concerning words for body parts. In fact, some southern variants, including Antandroy, have an ordinary word and a respectful one for body parts, but only in the Androy region the respectful words are of daily use (see [74] for an exhaustive investigation into the origins of Malagasy terms for body parts). This is why our policy of choosing the most employed word for a given meaning resulted in some respectful words recorded in the Antandroy lists. If by brute force we substitute the polite forms with the common ones, the Antandroy isolation decreases and the three variants get closer to the Mahafaly and Antanalana variants. Nevertheless, while reduced, the isolation remains remarkable.

The Antaisaka and Zafisoro variants (violet) cluster with the blue group, but we decided to assign a different color because they are slightly isolated from the other blue dialects, possibly because they are transitional toward Central and Eastern dialects.

The Center-North-East branch, whose leaves are associated to green, red and orange, also splits in two sub-branches, a green one for Eastern and Central variants and a red one for Northern dialects. Sakalava of Mahajanga (orange) is transitional, being a Northern dialect (Sakalava Boina) it is influenced by Central varieties (especially Merina, the official language) and also Western dialects (Sakalava Menabe).

There is a strict correspondence between the UPGMA cladogram and geography, this can be easily perceived comparing colors in Figs 1 and 2.

The cladogram resulting from NJ algorithm is reported in Fig 3. NJ substantially gives the same cladistic results, the only difference being that the Antandroy variants remain even more isolated from the others and that Sakalava of Mahajanga more decidedly clusters with Northern dialects.

Both UPGMA and NJ rely on the assumption that transmission is mostly vertical, so that a phylogenetic cladogram is fully justified. The main theoretical difference between the two algorithms is that UPGMA assumes that evolutionary rates are the same on all branches of the tree, while NJ allows differences. The question of which method is best suited to infer the phylogeny has been well studied and the general consensus is that NJ usually comes closer to the true cladistic (if it exists and it is mostly generated by philogenesys). Nevertheless, UPGMA more easily can be used to attribute dates since it assumes that the distance from the root is the same for all leaves, performing, in practice, a kind of average over different evolutionary rates. Luckily, in the present case no significant qualitative differences separate the two cladograms.

However, in Section 4 we show that relations among dialects are not necessarily tree-like, and we propose a totally different method for cladistic which privilege horizontal instead vertical transmission. Unexpectedly, the resulting cladogram is completely compatible with those in this section.

Finally, it is interesting to remark that the qualitative aspects of the cladograms are dominated by the least stable items, *i.e.*, those items whose corresponding words undergo to a faster evolution (more often replaced or modified). This is not astonishing since many items are designated by identical or very similar words in all dialects so that they are totally useless for any kind of cladistic. By definition, these are the most stable items, while the least stable items are those designated by different words in different dialects so that they are very useful for cladistic.

We generated two UPGMA trees, the first (S1 Fig) from Swadesh lists with the 35 most stable terms, and the second (S2 Fig) from Swadesh lists with the 35 least stable terms. It can be observed that while the first tree is totally different from the one of Fig 2, the second is very similar.

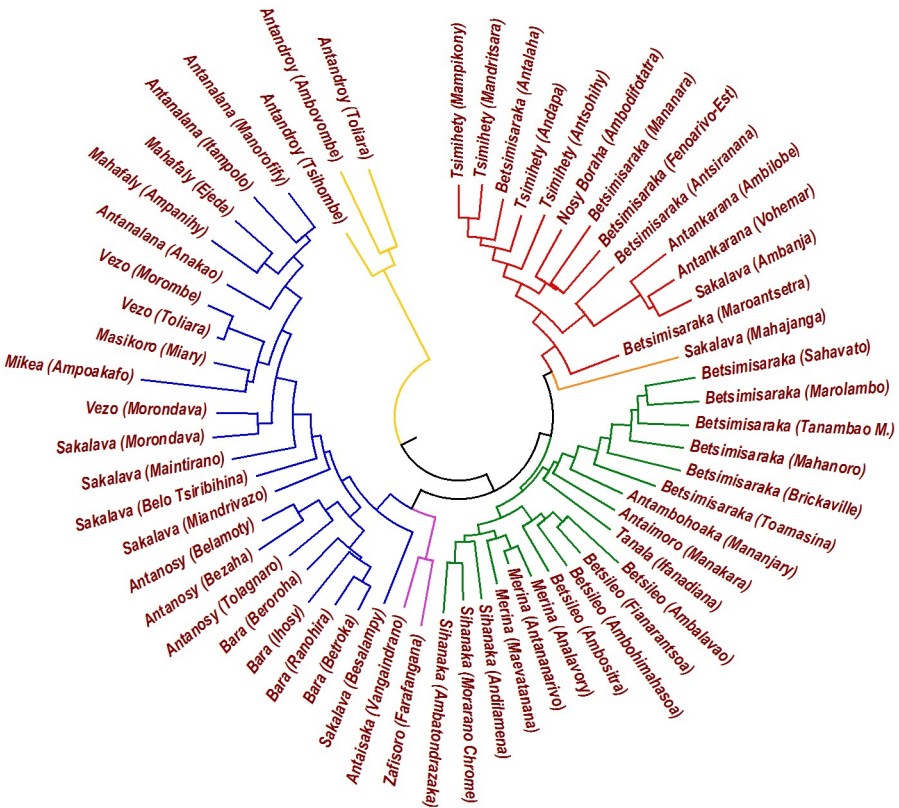

**Fig 3. NJ tree obtained from the genealogical distances.** The classification is the same of the UPGMA tree, the only difference being that the Antandroy variants remain even more isolated from the others and that the Sakalava of Mahajanga more decidedly clusters with Northern dialects.

Notice that this instance is reversed when one has to compare languages with a very remote common ancestor, in this case, in fact, only the extremely stable items are useful [71].

## 3.2 A date for the arrival of the ancestors of modern Malagasy people

The date of the landing of the Asian ancestors of nowadays Malagasy people or, more precisely, the date of the start of the diversification of Malagasy varieties, is fixed by choosing $\tau$. For example, the choice of $\tau$ in Fig 2 determines the genealogical distance between the root and the leaves, this distance is by definition the time that separates all contemporary Malagasy dialects from their last common progenitor (Proto-Malagasy).

As already mentioned, in the idea of Swadesh $\tau$ should have been a universal constant but this is not the case for many reasons, as the incidence of horizontal transfers and the differences in replacement rates for different meaning or different language families. To overcome these difficulties Swadesh himself proposed for $\tau$ a value smaller than the inverse of the replacement rate but without a clear recipe for fixing its actual value.

In principle one can fix $\tau$ by a single known historical date. For example, Icelandic (language 1) and Norwegian (language 2) started to separate $T(1, 2) = 1100$ years ago; on the other side their lexical distance $D(1, 2)$ or their overlap $C(1, 2) = 1 - D(1, 2)$ can be easily computed from vocabulary by Swadesh approach; then, from the equation $T(1, 2) = -\frac{\tau}{2} \ln C(1, 2)$ the value of $\tau$ remains determined because both $T(1, 2)$ and $D(1, 2)$ are known.

This method is not precise since the computed $\tau$ may be different for different pairs of languages in the same family and, in general, it is different for different families.

We propose here a strategy which is also based on history, but instead of considering a single pair of languages, we perform a kind of average over all the genealogical distances between each member of a family and their last common ancestor. The present approach is totally different and much more intuitive with respect to the one which was used in [13], nevertheless the result is the same: the landing of the Asian ancestors occurred around 650 *CE*.

To reach our goal we simply need a family whose last common ancestor is known and well temporally localized. For our purposes we found that a good choice was to consider the family composed by 46 Romance languages spoken in the geographically continuous territory represented the Italian peninsula, Iberian peninsula and Southern Gaul (see Fig 4 for the names of the languages). We chose this territory because all dates are known and because from the end of its Roman colonization to present, the language dynamics in this region was a mix of

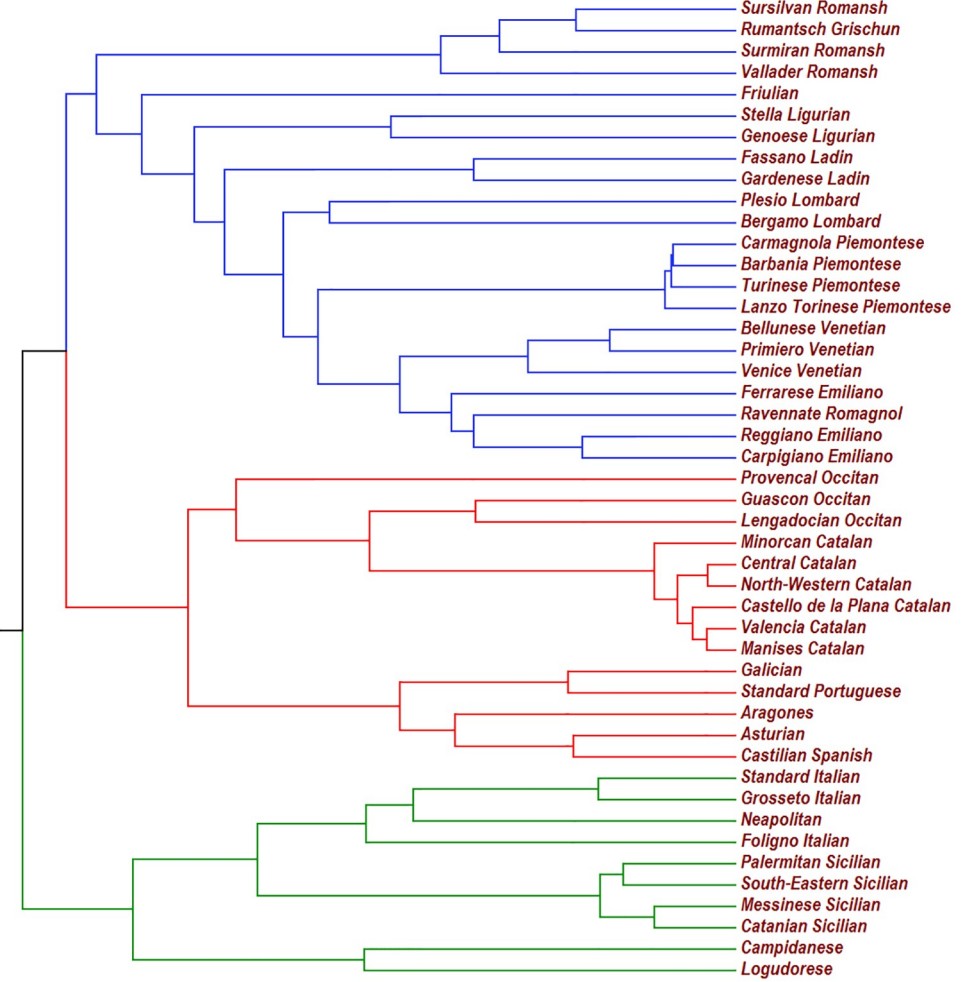

**Fig 4. This UPGMA tree is obtained from the 1035 genealogical distances** $T(\alpha, \beta) = -\frac{\tau}{2} \ln C(\alpha, \beta)$ **between all pairs which can formed by 46 Romance languages spoken in the Italian peninsula, Iberian peninsula and Southern Gaul.** According to our evaluation these Romance languages started to separate each other between 2121 and 2033 years *BP*. This range for the genealogical distance from the root to the leaves implies $5027 \leq \tau \leq 5245$. Notice that the main partition corresponds to the La Spezia–Rimini Line (also known as the Massa–Senigallia Line), which always seems quite mysterious for Italian native people.

phylogenesys and horizontal transfers between geographically contiguous languages without main borrowings from non-Romance languages. In other words, a dynamics very similar to the Malagasy one.

After having determined the overlap $C(\alpha, \beta)$ between all the $46 \times 45/2 = 1035$ pairs of languages in this family (from the dataset described in previous section), we determine the genealogical distances $T(\alpha, \beta) = -\frac{\tau}{2} \ln C(\alpha, \beta)$. Then, we built a UPGMA tree (Fig 4) and we get a value $T = 0.4044 \times \tau$ for the genealogical distance from the root to the leaves. If one knows from history that these 46 Romance languages started to differentiate $T$ years *BP* one immediately gets $\tau = T/0.4044$.

In order to evaluate $T$, the relevant events are:

- 102—101 *BCE*—The Teutons and the Cimbri are defeated in the battles of Aquae Sextiae and Vercellae. The Roman dominion over Iberian Peninsula and Southern Gaul (Occitania) is undisputed and definitely consolidated. More the 90% of the territory is under Roman control.

- 19 *BCE*—The Astures and the Cantabri surrender to Rome, ending the Cantabric Wars. The Iberian peninsula is totally and definitively under the Roman rule.

- 24—14 *BCE*—The army sent by Caesar Augustus beats the last Ligurian tribes who still resisted Roman domination. The Roman conquest of Southern Gaul (Occitania) is completed.

According to these events, the 46 Romance languages that we considered started to separate between 2121 and 2033 years *BP*. From $\tau = T/0.4044$ we get the range $2033/0.4044 = 5027 \leq \tau \leq 2121/0.4044 = 5245$.

In turn, this range of values for $\tau$, when used for the UPGMA tree constructed from the genealogical distances of Malagasy variants (Fig 2), gives a value between 1338 ($\tau = 5027$) and 1396 ($\tau = 5245$) for the genealogical distance from the root to the leaves. This means a start for the diversification of Malagasy variants between $2019 - 1396 = 623$ *CE* and $2019 - 1338 = 681$ *CE*. In conclusion, it can be stated that the Asian ancestors of nowadays Malagasy people landed around 650 *CE* or shortly before.

## 3.3 Who was there before?

Malagasy mythology portrays a people, called the Vazimba, as the original inhabitants, preceding the arrival of the ancestors of modern Malagasy (but this term may also refer to the people ruled by the first kingdoms which established on the highlands, not a different people from nowadays Malagasy).

Archaeology tried to carry this belief on a scientific ground answering the question: when first humans reached Madagascar? We have seen that the answer is not univocal, with some authors claiming a very early presence dating thousands of years, and other authors which deny any presence before the second half of the first millennium *CE*.

Since linguistics and also genetics point for a date of arrival of the ancestors of modern Malagasy around the seventh century *CE*, the question for linguistics and genetics becomes: was the Island populated by a previously arrived people which possibly left tenuous traces in some Malagasy vocabularies or some unique genetic marks?

Some authors tried to find these linguistic traces. In [25] it is argued that ". . . low-density hunting-gathering populations probably did cross the Mozambique Channel and begin to exploit the Malagasy environment. Such populations would probably have been physically like the present-day Hadza of Tanzania rather than Khoesan speakers. It is likely that these survive

in the present-day Mikea/Vazimba populations; although today they speak Malagasy dialects, there are clear cultural and linguistic traces of a distinct origin".

Moreover, in [75] the authors write "Generally speaking, the Vazimba data is so exiguous and so scattered, that it is unlikely to constitute a significant record of a substrate vocabulary, as opposed to a set of idiosyncratic words, not untypical for a population of former foragers. However, [...] it looks as if there may be genuine substrate vocabulary in Beosi and that this could reflect the speech of a forager group which migrated from the African mainland in pre-Austronesian times."

Indeed, these papers fail in providing any evidence, moreover they superficially individuate the subject of their research "Scattered among the Malagasy live groups of hunter-gatherers variously known as the Mikea or Vazimba". In reality Mikea are not a constellation of tribes scattered all around but a specific population living in the Mikea forest, North of Toliara. One of the authors of the present paper (M.S.) inhabited in the area for few months, moreover, many scholars dedicated long studies to this people (see for example [76, 77]).

Furthermore, the authors of [78] after having performed a genomic study over three population including Mikea declared "We were unable to detect in Mikea unique ancestry components that would have been absent among sampled populations. These results support the hypothesis that the Mikea originated from agricultural populations and have reverted to the forest. Despite the Mikea population deriving from the same genetic admixture as Vezo and Temoro, they have reverted to a hunter-gatherer lifestyle."

In conclusion, evidence points to a phenomenon of cultural reversion to a hunter-gatherer life style of a small population [76], which is not the only case ever observed [79].

Analyzing the point with the help of the two trees that we have generated (Figs 2 and 3), the Mikea variety is well collocated in the Southern blue group. In the UPGMA tree it is scattered to form a separated branch, but in the NJ tree it finds its position close to the neighboring Vezo and Masikoro dialects. Since NJ is usually able to detect deeper relationships between languages than UPGMA, we argue that the relative isolation in UPGMA is due to the recent drift after Mikea embraced a Paleolithic lifestyle.

We also measured the average lexical distance of each language from the others, finding that this quantity is large for Mikea, but not larger than those we found for Antandroy dialects (S3 Fig).

Finally, in the analysis of next section where languages form two-dimensional patterns, again we find that Mikea variety clearly belongs to the Southern blue cluster, even if it is slightly scattered away from the center of it.

In conclusion, our opinion is that Mikea is a standard Malagasy variety. Differences, which are likely a consequence of recent isolation due to their peculiar lifestyle, seem to be the result of some modern inventions, probably a strategy to self-protect from external intrusions.

## 4 Reconstructed geography

Although genealogical descriptions are ubiquitous in linguistics, they can fail in recovering and representing some information contained in the matrix of genealogical distances. The main reason of this inadequacy lies in the horizontal transfer process between geographically close languages (vocabulary borrowings). In lexicostatistics borrowings are treated as a sort of annoying noise which dirties the "real" vertical process and which can be eventually eliminated or neutralized by a careful screening performed by experts.

In reality, things are quite different. When a family of languages represents a continuum both geographically and linguistically as, for example, Romance family of languages or Malagasy family of varieties, horizontal transfers turn out to be a primary aspect of the dynamics of

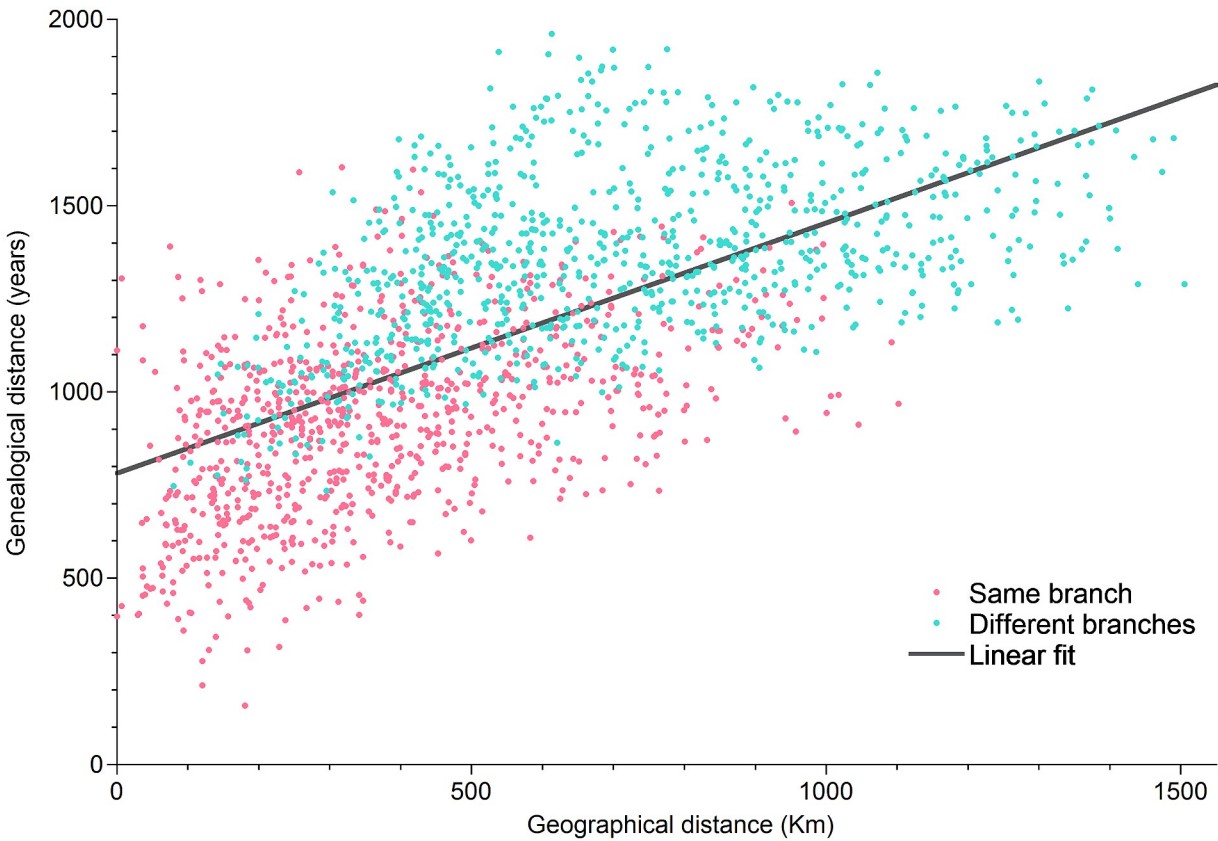

**Fig 5. Genealogical distances $T(\alpha, \beta)$ against geographical distances $K(\alpha, \beta)$ (great-circle distances in Km between towns).** Correlation is 0.65 which is quite high indicating that geography is complementary to genealogy. This will be useful for the definition of Diversity. The parameters of the linear fit $T = a + bK$ are: $a = 78 \cdot 10$ years and $b = 0.67$ years/Km. Light green points represent pairs of dialects in which one of the two belongs to the South-West branch and the other to the Center-North-East one, while light red ones correspond to pairs in which both belong to the same main branch. Given the same geographical distance, if two towns belong to different branches they are likely more genealogically distant (light green points are mostly above the light red ones when co-present at a given geographical distance).

languages [72]. The horizontal contacts break the purely ultra-metric phylogenetic structure of the matrix of genealogical distances, thus, in the translation of the matrix information into a phylogenetic tree, some of it can be totally lost. In general, the lost information concerns geography which is relevant both in linguistics and in biology.

Fig 5 shows that there is a strong linear 0.65 correlation between genealogical distances and geographical distances. This means that geography cannot be ignored since horizontal transfers weave an affinity network which connects geographically close dialects [80].

On the other side, this does not mean that geography explains everything, in fact, the fundamental role of vertical transmission is also reflected in Fig 5. Light green points represent pairs of dialects in which one of the two belongs to the South-West branch and the other to the Center-North-East one, while light red ones correspond to pairs in which both belong to the same main branch. Given the same geographical distance, if two towns belong to different branches they are likely more genealogically distant (light green points are mostly above the light red ones when co-present at a given geographical distance).

In order to grasp the complementary information related to the horizontal dynamics, we revisit a method proposed in [80]. This method allows to reconstruct the geography of Madagascar by the matrix of genealogical distances. The reconstruction would be perfect if all points

in Fig 5 were lying on a line; this is obviously not the case, nevertheless we succeed in an approximate reconstruction which allows to draw some further conclusions concerning the relationships among Malagasy dialects.

The strategy only uses genealogical distances as input, no hints come from the geography, *i. e.*, by the "true" geodesic coordinates of the towns/villages.

We first introduce a cost function *R* defined as

$$R(\mathbf{x}_1, \ldots, \mathbf{x}_N) = \frac{2}{N(N-1)} \sum_{\alpha,\beta} |T(\alpha,\beta) - |\mathbf{x}_\alpha - \mathbf{x}_\beta||. \quad (5)$$

where the sum goes on all $N(N-1)/2 = 1770$ possible pairs.

The $N = 60$ two-dimensional coordinates $\mathbf{x}_\alpha = (x_\alpha, y_\alpha)$ represent the positions of the $N$ towns/villages on a fictitious plane where coordinates are measured in years. Prompted by the correlation in Fig 5, we can say that the optimal positions $\bar{\mathbf{x}}_1, \bar{\mathbf{x}}_2, \ldots, \bar{\mathbf{x}}_N$ of the $N$ towns/villages are those that minimize the cost function:

$$R_{min} = \min R(\mathbf{x}_1, \ldots, \mathbf{x}_N) = R(\bar{\mathbf{x}}_1, \ldots, \bar{\mathbf{x}}_N). \quad (6)$$

The purpose of the minimization is to classify the dialects according to their genealogical distances, the more these distances have a two-dimensional structure (as, for example, the geographical distances over a map), the more the recontruction is accurate.

Notices that the minimum is not unique since a rotation and/or a translation and/or an overturning of the plane leave unchanged the distances $|\bar{\mathbf{x}}_\alpha - \bar{\mathbf{x}}_\beta|$.

Numerical minimization is not as easy as it looks like, we are forced to proceed in a iterative manner. Before starting minimization, we rank all dialects from 1 to 60 according to the decreasing value of $\Sigma_\beta T(\alpha, \beta)$. All positions except for $\alpha = 2$ and $\alpha = 3$ are initially fixed in the origin. The second town ($\alpha = 2$) is fixed in one of the infinite positions at the correct distance $T(1, 2)$ from the first town ($\alpha = 1$), and the third town ($\alpha = 3$) is fixed in one of the two points which are at the correct distance $T(1, 3)$ from the first town and at the correct distance $T(2, 3)$ from the second one. We proceed in this way both to break the symmetry of the initial condition with all towns in the origin which is a local minimum and to make a choice among all possible equivalent minima. Obviously, any other choice for initial condition which breaks the symmetry is equivalent.

Then, following the ranking, we minimize *R* step by step, any time with respect to a single two-component variable $\mathbf{x}_\alpha$, proceeding from $\alpha = 1$ to $\alpha = 60$. We repeat this procedure cyclically, restarting every time from the first dialect and following the ranking. We go on until we observe a whole loop in which the function *R* stops decreasing. Eventually, we raise the precision of our numerical research, and we continue the minimization until a steady point is reached again.

The optimal positions output $\bar{\mathbf{x}}_1, \ldots, \bar{\mathbf{x}}_N$ is a set of $N = 120$ scalar values chosen in order that the $N(N-1)/2 = 1770$ metric distances $|\bar{\mathbf{x}}_\alpha - \bar{\mathbf{x}}_\beta|$ best represent the corresponding 1770 genealogical distances $T(\alpha, \beta)$. This representation cannot be perfect given the difference between the two numbers: 120 and 1770.

We stress again that we do not use any information coming from real geography, *i.e.*, we don't use the geodesic coordinates of town/villages. Using a colorful picture, we could say that one could have an idea of Malagasy geography after a discussion about European Rugby or American movies with people from different regions of the Island.

For visual comparison of the virtual positions $\bar{\mathbf{x}}_\alpha$ with the "true" geodesic coordinates, we overturn, rotate, translate and zoom the plane with all the $\bar{\mathbf{x}}_\alpha$ before positioning it on chart with the profile of Madagascar. This operation only has the goal to make our result more

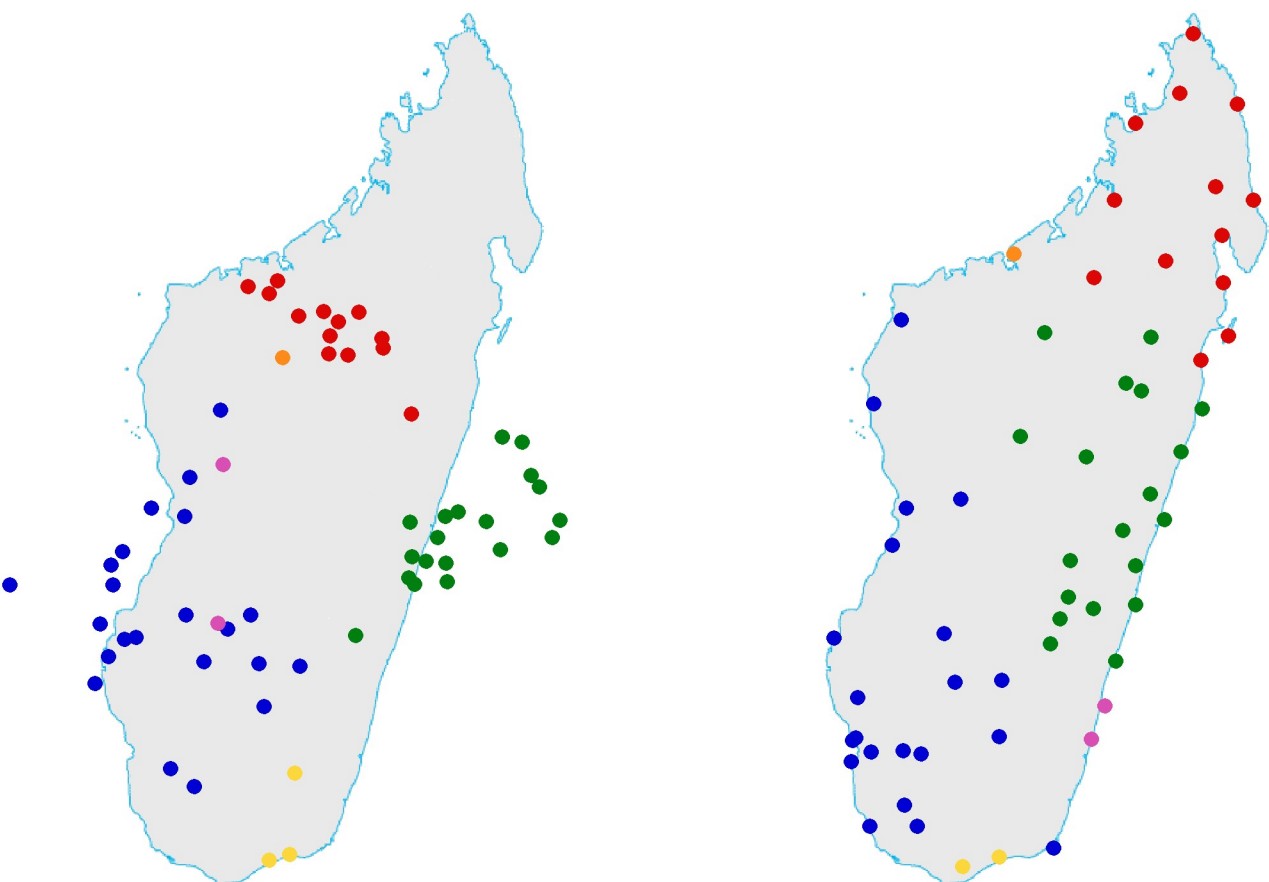

**Fig 6. At the left the chart of Madagascar reconstructed by lexical input only, at the right the "true" chart.** There are three yellow spots in the left chart but only two in the right one because Antandroy (yellow) and Vezo (blue) are superposed in Toliara. To describe the result in this figure by a colorful picture, we could say that the chart at the left is the idea one could have of Malagasy geography after a discussion about European Rugby or American movies with people from different regions of the Island.

readable, but in no way is it necessary. The result can be appreciated in Fig 6, where the reconstructed geography is on the left and the real geography on the right. Given that the left chart is obtained only from vocabularies, the correspondence is surprising.

The clustering of dialects is the same of the trees (we have maintained the same colors), but with some new details. First, Northern dialects (red) are very close each other as well the Central-Eastern dialects. This lack of diversity may indicate that these regions were reached after the Southern regions were already colonized. Second, Mahajanga undoubtedly groups with Northern dialects, confirming that the people of Mahajanga belong to the Boina branch of Sakalava ethnicity. Third, the Antaisaka and Zafisoro dialects (violet) decidedly group with South-Western dialects despite their transitional geographical position on the East coast. Last, the blue spot in Mozambique Channel refers to Mikea people. Once more it seems that they well cluster with the other South-Western dialects even if they drifted in a relative isolation probably because of some recent linguistic evolution.

In order to quantitatively compare a tree representation with the geographical one, we consider the reached minimum $R_{min} = \frac{2}{N(N-1)} \sum_{\alpha,\beta} |T(\alpha,\beta) - |\bar{\mathbf{x}}_\alpha - \bar{\mathbf{x}}_\beta||$ and the analogous $R_u = \frac{2}{N(N-1)} \sum_{\alpha,\beta} |T(\alpha,\beta) - T_u(\alpha,\beta)|$ where the $T_u(\alpha,\beta)$ are the genealogical distances as reconstructed by UPGMA. In the first case the value is 194, in the second 205, meaning that the two

complementary description are almost equally accurate, with a negligible advantage for the geographical one. Given that the average $\frac{2}{N(N-1)}\sum_{\alpha,\beta}T(\alpha,\beta)$ of the genealogical distances equals 1148, both procedures account for more than 80% of its value.

Finally, we acknowledge that our strategy operates the reconstruction of geography on a plane, while Madagascar occupy a portion of the surface of a sphere. Nevertheless, the approximation is extremely accurate so that the effort of using a set of geodesic fictitious variables would not be justified.

## 5 Diversity: Landing and settlement

The center of dispersal of Malagasy variants, likely coinciding with the landing spot, can be inferred by finding the site with largest Diversity. The idea that the homeland of a biological species or a language group corresponds to the region with the greatest Diversity was proposed about one century ago in biology [32] and in linguistics [31], and it is widely accepted.

More recently, this idea was transformed into quantifiable terms by comparing linguistic and geographical distances [30], and it was later adapted to determine the center of dispersal of Malagasy variants [13].

There are two residual problems with this quantitative approach, the first is that Diversity was defined only in towns/villages where the dialects were collected, and the second is that Diversity was not defined as a local quantity, *i.e.*, distant towns contributed as much as close ones to its makeup. Here we modify the definition in order that: i) Diversity remains determined in every site of the Island; ii) its degree of locality can be tuned.

In Fig 5 we plotted the genealogical distances $T(\alpha,\beta)$ against the geographical distances $K(\alpha,\beta)$ (the great-circle distances in Km between towns/villages). Correlation is 0.65 which is quite high and which implies that genealogical distances increases along with geographical distances. The linear fit $T(\alpha,\beta) = a + bK(\alpha,\beta)$ has parameters $a = 78 \cdot 10$ years and $b = 0.67$ years/Km. If all points were on the line, one would have $\frac{T(\alpha,\beta)}{a+bK(\alpha,\beta)} = 1$ for each pair $\alpha$ and $\beta$. In reality, for some points (for some towns/villages) this ratio is larger than one (those above the line), and for others is smaller (those below the line).

The points above/below the line have a genealogical distance that is larger/smaller of what is explained in terms of geographical distance, therefore, they are more/less lexically diverse than expected. Prompted by this analysis, we define the Diversity between two towns as the ratio

$$V(\alpha,\beta) = \frac{T(\alpha,\beta)}{a+bK(\alpha,\beta)}. \tag{7}$$

Eventually, one could define the diversity in town village $\alpha$ as the average over $\beta$ of this quantity, apart from details, this was the proposal in [30] and [13]. Unfortunately, this quantity is defined only in towns/villages and it is non-local (distant towns/villages contribute as close ones to its makeup).

Here we would like to define a local Diversity in any geographical site of the Island whose geodesic coordinates are indicated by $\boldsymbol{\xi}$. This definition has to be an average of all $V(\alpha,\beta)$ weighted in order that the more the towns/villages are close to $\boldsymbol{\xi}$, the more the corresponding $V(\alpha,\beta)$ contributes. A natural choice is

$$V(\boldsymbol{\xi}) = \frac{\sum_{\alpha,\beta}e^{-\epsilon\Delta(\xi,\alpha,\beta)}V(\alpha,\beta)}{\sum_{\alpha,\beta}e^{-\epsilon\Delta(\xi,\alpha,\beta)}} \tag{8}$$

where the sums go over all the $N(N-1)/2$ pairs of towns/villages, and

$$\Delta(\xi, \alpha, \beta) = K(\xi, \alpha) + K(\xi, \beta) \tag{9}$$

and where $K(\xi, \alpha)$ and $K(\xi, \beta)$ are the great-circle distances in Km between a generic point of the Island with geodesic coordinates $\xi$ and, respectively, the towns/villages $\alpha$ and $\beta$. The proximity parameter $\epsilon$ localizes the Diversity implying a cutoff for those pairs of town/villages whose $\Delta(\xi, \alpha, \beta)$ exceeds $1/\epsilon$. Roughly speaking, towns whose geographical distance from $\xi$ is less than $1/2\epsilon$ most contribute to the value of the Diversity in $\xi$.

Fig 7 shows the Diversity $V(\xi)$ corresponding to the choice $\epsilon = 0.003$ which implies the characteristic scale $1/2\epsilon = 167 Km$ (different choices lead to similar qualitative results). The figure displays a very clear reduction of Diversity going from South or South-East to North. In particular, the extreme North exhibits a very low diversity, which is coherent with the chart at the right in Fig 6 (geography reconstructed by lexical data), where Northern towns/villages appear extremely close each other.

The conclusion is that the best candidate for the homeland is the South-East coast. The Northern locations are the least diverse and they must have been settled last. This is the same result found in [13].

The identification of the South-Eastern coast of Madagascar as the landing spot of the Asian ancestors of nowadays Malagasy people is corroborated by other observations. One of the major currents in the Indian Ocean is the South Equatorial Current that goes from Sumatra to Madagascar. When Mount Krakatoa erupted in 1883, pumice was transported to the east coast of Madagascar, where the Mananjary river empties into the sea (between Farafangana and Mahanoro). During the Second World War pieces of wreckage from ships sailing between Java and Sumatra which had been bombed by the Japanese air force also arrived in South-Eastern coast [81].

According to these facts, the ancestors of nowadays Malagasy people probably passed by the easily navigable Sunda strait and, with the help of the South Equatorial Current, they reached, intentionally or unintentionally, the South-East coast of Madagascar. The South-East landing spot hypotheses also adds fuel to the all-in-one-voyage conjecture, since an intermediate stopover in the East African coast would have more likely implied landing on the North-Western coast. This location is also coherent with [47], where it is found that a very small number of women from Indonesia contributed to the genetic pool of the founding population, probably as a consequence of an unintentional colonization event.

In [34], it is argued that the distribution of Diversity is an artefact due to continued contact with Indonesia for centuries after the first landing, this would explain the high Southern Diversity. Adelaar also argued that the Asian ancestors of Malagasy were following established trade routes and were interested in trade contacts: they were not passively following sea currents. Furthermore, according to Adelaar, the relatively homogenous Malagasy language with Bantu core elements, most likely is due to a formation outside Madagascar followed by a bottleneck at the arrival on the island.

Although we cannot exclude this possibility, we think that the scenario that Fig 7 gives us back is more easily explained by a progressive colonization from South to North rather than by long-standing post-migratory influences. Moreover, it is not clear how continued contact in the South could be responsible for the severe reduction in diversity in the far North. Nevertheless, we are also aware that our scenario is also hypothetical, and we think that more investigation is needed to reach a consensus on this important piece of the history of the colonization of Madagascar.

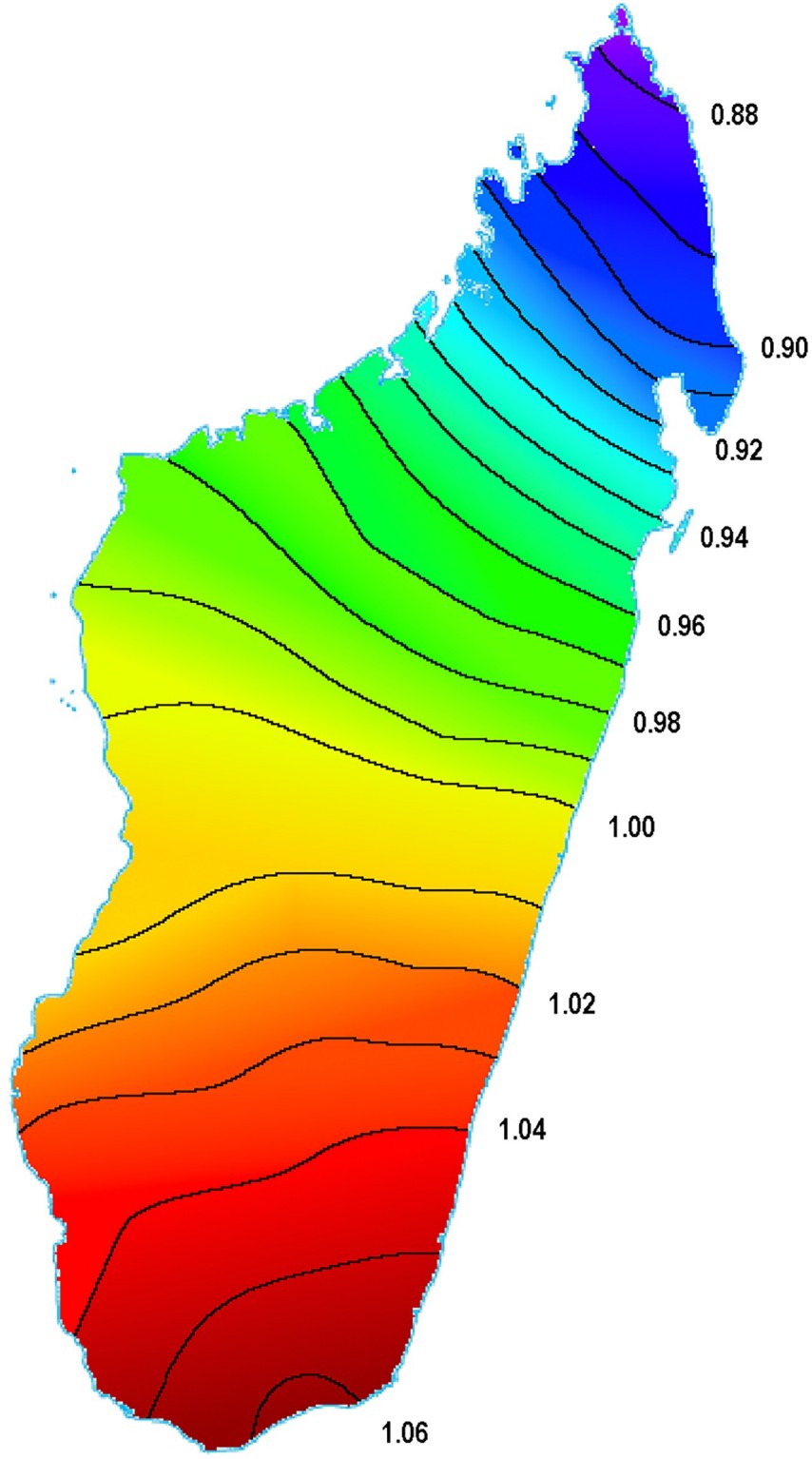

**Fig 7. The figure shows a progressive reduction of Diversity going from South or South-East to North.** The extreme North sees a severe reduction of Diversity. The value of the proximity parameter is $\epsilon = 0.003$, but other choices lead to qualitatively similar results. This picture points to South or to South-East as the dispersal center of Malagasy variants and, consequently, as the landing spot of the Asian ancestors.

## 6 Conclusions and outlook

All results in this paper rely on two main ingredients: a new dataset of 207 words Swadesh lists for 60 different variants of the Malagasy language, and on the construction of a matrix of genealogical distances between pairs of dialects. This new dataset overall 12,420 terms, is by far the best available, both for dimension and completeness and, consequently, the matrix of distances is the largest ever constructed. Then, the content and the findings can be summarized as follows:

- the distances are classified through two different types of phylogenetic algorithm (NJ and UPGMA). Both cladograms show that the family of dialects splits into two main branches: South-West and Center-North-East. In turn, the Center-North-East branch splits in a North and in Center-East sub-branches while the South-West branch sees Antandroy variants well separated from the others. These subdivisions are robust and coincide with the analysis performed in [13] based upon a different dataset. The precise collocation of a variety (Sakalava of Mahajanga) remains poorly determined, as well as it is not completely clear if Zafisoro and Antaisaka varieties should be considered transitional or simply Southern dialects;

- the landing date of the ancestors of Malagasy is determined around 650 *CE*. This result is obtained by a new approach based on the comparison of Malagasy family tree with the similar tree of a Romance family which includes the languages of the the Italian peninsula, the Iberian peninsula and Southern Gaul (Occitania). For this family all dating are well historically attested and this knowledge is used to measure the deepness of the Malagasy family of dialects. Our 650 *CE* dating is shared by many studies concerning Madagascar including [13] where it is obtained by a completely different methodology;

- the Malagasy mythology portrays a people, called the Vazimba, as the original inhabitants. Some authors argued that the aboriginal vocabulary could have left traces into the dialects spoken by the residual ethnicities of hunter-gatherers. Mikea dialect, spoken by a few thousand people living in the homonymous forest in the South-West of Madagascar, was considered one of the most promising candidates. Our analysis concludes that Mikea variant is not very different from their neighbors Vezo and Masikoro variants, eventually showing some innovation due to the long isolation and to the will of preserving the identity;

- the matrix of genealogical distances, beyond phylogenetic relationships, also contains the information concerning horizontal dynamics, *i.e.*, the process of vocabulary borrowings between geographically close regions. We revisit a method proposed in [80] which allows to grasp this information and to infer the geography of Madagascar only by lexical comparisons. The resulting two-dimensional chart leads us to draw some supplementary conclusions: the Sakalava variety of Mahajanga belongs to the Northern group (Sakalava Boina) while Zafisoro and Antaisaka varieties are Southern dialects;

- the center of dispersal of Malagasy variants can be inferred by finding the site with the largest Diversity. In this paper we propose a new definition of Diversity which has a value in every site of Madagascar (not only in the sites where the vocabulary was collected), and which is locally defined (in each site only depends on nearby measurements). We find out that the center of dispersal of Malagasy variants (likely the landing spot of the Asian colonizer) is on the South-East coast. The South-East landing spot hypotheses also adds fuel to the all-in-one-voyage conjecture, since an intermediate stopover in the East African coast would have more likely implied landing on the North-Western coast. Nevertheless, we akwnoledge that the Southern high diversity could be the consequence of later contacts with Indonesia. More investgation is needed.

In future research we plan to address two old but still debated topics concerning the relationships of Malagasy with Indonesian and African languages. The novelty of our approach should be considering Malagasy as a constellation of dialects, rather than a single language, while comparing it with other languages.

First, we would like to verify the statement contained in [26] "[. . .] the older borrowings seem to have sources in Swahili and precursors of Swahili and not in a scatter of coastal Bantu languages as might be expected. In particular, there seems to be no particular link with Mozambican languages". Preliminary research seems to point in the contrary direction. It would be interesting to have more information on the relations with specific Bantu languages.

We also would like to give a new look to the relations of Malagasy people and language with Eastern Indonesia and, eventually, with sea nomads populations, in the aim to fill the gap between Biology and Ethnology on one side and Linguistics on the other (musical instruments, Polynesian motif). This research could also provide some element for the single/multiple colonization debate.

## Supporting information

**S1 Dataset. Malagasy Swadesh lists.** The complete dataset of 207 items Swadesh lists for 60 Malagasy variants in text format. The entire 12,420 items dataset which can be consulted here can be freely used, as long as its origin is quoted.
(PDF)

**S1 Table. Ethnicities, towns and coordinates.** The name of the dialects (ethnicities), the name of the Towns/Villages where dialects were collected together with their coordinates. Latitude and longitude are given in degrees complete of decimals, therefore, the two digits after point do not represent seconds but degree's decimals.
(PDF)

**S2 Table. Lexical distances.** This table contains the $N \times N$ upper triangular matrix whose entries are the $N(N - 1)/2 = 1770$ lexical distances $D(\alpha, \beta)$ between all pairs of languages.
(PDF)

**S1 Fig. UPGMA tree by the most stable items.** The UPGMA tree generated by the Swadesh lists which only contain the 35 most stable items for the family of Malagasy variants.
(PNG)

**S2 Fig. UPGMA tree by the least stable items.** The UPGMA tree generated by the Swadesh lists which only contain the 35 most stable items for the family of Malagasy variants.
(PNG)

**S3 Fig. Average lexical distances.** The average lexical distance of each dialect from the others. The 60 varieties are ranked according to the value of the average.
(PNG)

## Acknowledgments

We are grateful to the Professors Heriniaina Andry Raboanary, Toky Hajatiana Raboanary, Julien Amédée Raboanary, Mara Edouard Remanevy, Dimby Vaovolo, Barthélemy Manjakahery and Marius Mandimbitafika Sambizafy for their invaluable advice, suggestions and data supply.

We are also grateful to Alexander Adelaar, Murray Cox, Davide Vergni, Angelo Vulpiani, Karl Eggert, Filippo Petroni, Rory Van Tuyl and Dimitri Volchenkov for a critical reading of the manuscript, suggestions and criticism.

The research has benefited from the logistical support of the Institut Supérieur Polytechnique de Madagascar (ISPM) and the invaluable assistance of its teachers and students.

## Author Contributions

**Conceptualization:** Maurizio Serva.

**Data curation:** Maurizio Serva.

**Formal analysis:** Michele Pasquini.

**Investigation:** Maurizio Serva.

**Software:** Michele Pasquini.

**Supervision:** Maurizio Serva.

**Writing – original draft:** Maurizio Serva.

**Writing – review & editing:** Michele Pasquini.

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
