## [Decision Letter · Decision Letter 0]

22 Sep 2020

Dialects of Madagascar

PONE-D-20-15408

Dear Dr. Pasquini,

We’re pleased to inform you that your manuscript has been judged scientifically suitable for publication and will be formally accepted for publication once it meets all outstanding technical requirements.

Kind regards,

Enrico Scalas, Ph.D.

Academic Editor

PLOS ONE

Additional Editor Comments (optional):

Both referees are very positive and it has been a pleasure to act as AE of your paper in a difficult period for everybody.

1. PLOS ONE requires titles be specific, descriptive, concise, and comprehensible to readers outside the field In this case we have concerns the title may not be suffficiently descriptive of the research that was conducted https://journals.plos.org/plosone/s/submission-guidelines#loc-title

Reviewers' comments:

Reviewer's Responses to Questions

**Comments to the Author**

1. Is the manuscript technically sound, and do the data support the conclusions?

Reviewer #1: Yes

Reviewer #2: Yes

2. Has the statistical analysis been performed appropriately and rigorously? 

Reviewer #1: Yes

Reviewer #2: Yes

3. Have the authors made all data underlying the findings in their manuscript fully available?

Reviewer #1: Yes

Reviewer #2: Yes

4. Is the manuscript presented in an intelligible fashion and written in standard English?

Reviewer #1: Yes

Reviewer #2: Yes

5. Review Comments to the Author

Reviewer #1: My identity can be revealed to the authors. They had already sent the manuscript to me previously and we had an in-depth discussion about its contents. I approve of the methodology and the handling of the linguistic data. I cannot comment on the statistics as I lack the necessary expertise.The manuscript is of a very sound quality. I agree with most of the authors' conclusions. On one particular issue (the region in Madagascar where Austronesian speakers first arrived) we agree to disagree. However, the authors clearly justify their position on this point, which is valuable as it will enable readers to appreciate the problem.

Reviewer #2: I first apologise with the authors for taking so long in providing

them with a report. I liked a lot the paper, and I recommend its

publication. I think the article represents a great piece of work from

several points of view. First of all, from the data perspective: the

authors put together an impressive set of Swadesh lists which

represents a unicum for Malagasy languages in the literature. Thanks

to this solid basis and their experience in phylogeny reconstruction,

the authors embarked in a journey to address fundamental questions

about the peopling of Madagascar, in terms of historical times and

initial location of the settlements, and the current phylogenetic

structure of present varieties of Malagasy. I think the results are

robust and convincing, and my impression is that this paper will stand

as a reference for future investigations about Madagascar and its

languages. I also appreciated the technical part of the approach and the

original solutions the authors conceived to overcome long-standing

problems in lexicostatistics and glottochronology. My only

advice is to take care of the language to make it smoother and less

redundant. At times the paper is unnecessarily verbose while I think a

good language sweep could make it sharper and ready for a broad

audience.

6. PLOS authors have the option to publish the peer review history of their article (what does this mean?). If published, this will include your full peer review and any attached files.

Reviewer #1: **Yes: **Alexander Adelaar

Reviewer #2: No

---

## [Editor Report · Acceptance letter]

24 Sep 2020

PONE-D-20-15408 

Dialects of Madagascar 

Dear Dr. Pasquini:

I'm pleased to inform you that your manuscript has been deemed suitable for publication in PLOS ONE. Congratulations! Your manuscript is now with our production department. 

Kind regards, 

on behalf of

Professor Enrico Scalas 

Academic Editor

PLOS ONE